# Negotiating Gestational Diabetes Mellitus in India: A National Approach

**DOI:** 10.3390/medicina57090942

**Published:** 2021-09-06

**Authors:** Uday Thanawala, Hema Divakar, Rajesh Jain, Mukesh M. Agarwal

**Affiliations:** 1Thanawala Maternity Home, Navi Mumbai 400701, India; udaythanawala@gmail.com; 2Divakar Specialty Hospital, Bengaluru 560078, India; drhemadivakar@gmail.com; 3Jain Hospital, Kanpur 208007, India; jainhospitals@gmail.com; 4Department of Pathology, California University of Science and Medicine, Colton, CA 92324, USA

**Keywords:** gestational diabetes, epidemiological studies, less-developed countries, prevention

## Abstract

The worldwide epidemic of diabetes mellitus and hyperglycemia in pregnancy (HIP) presents many challenges, some of which are country-specific. To address these specific problems, parochial resolutions are essential. In India, the government, by working in tandem with (a) national groups such as the Diabetes in Pregnancy Study Group of India, and (b) global organizations such as the International Diabetes Federation, has empowered the medical and paramedical staff throughout the country to manage HIP. Additionally, despite their academic university backgrounds, Indian health planners have provided practical guidelines for caregivers at the ground level, who look up to these experts for guidance. This multipronged process has helped to negotiate some of the multiple problems that are indigenous and exclusive to India. This review traces the Indian journey to manage and prevent HIP with simple, constructive, and pragmatic solutions.

## 1. Introduction

Although diabetes mellitus during pregnancy (DIP) causes hyperglycemia in pregnancy (HIP), it is gestational diabetes mellitus (GDM) that remains the major cause of HIP [1]. DIP, either antedating or detected during pregnancy, is the more hazardous form of HIP, producing severe hyperglycemia early in pregnancy, which persists postpartum; on the other hand, GDM causes mild hyperglycemia late in pregnancy, which usually disappears after delivery. As normal pregnancy advances, it causes insulin resistance. The resulting hyperglycemia is compensated for in healthy women by pancreatic beta cell hyperplasia, which can meet the additional metabolic demands. However, in GDM, there is an inadequate compensation, due to multiple genetic and environmental factors, causing hyperglycemia [2]. Though GDM is associated with maternal and fetal complications in index pregnancy, there are additional problems. After delivery, a woman with GDM is 10 times more likely to develop type 2 diabetes mellitus (T2DM) compared to a woman without GDM [3]. Intense screening, follow-up, and preventive measures for women with GDM after delivery can help to mitigate impending T2DM in an individual patient, and potentially in an entire population, epidemiologically [4]. Due to the current epidemic of T2DM worldwide, post-partum follow-up of women with GDM is crucial for all counties, especially countries with a high prevalence of diabetes mellitus (DM). India, being one such country, harbors the second highest number of adults with T2DM in the world, a number slated to increase by more than 75% in the next 25 years [5].

Therefore, it is critical for all countries to align themselves to the latest research on every aspect of GDM, from screening to management, and the critical long-term follow-up after delivery. The lack of a uniform global approach to GDM remains one major roadblock plaguing GDM and our prospects of turning the tide on the T2DM epidemic [6,7].

The focus of this article is to address the challenges, problems, solutions, opportunities, failures, and successes in managing GDM in south-east Asia (SEA), as exemplified by India, to stress how the unified effort of one country can battle the scourge of HIP.

## 2. Materials and Methods

Since the aim of this paper was to provide a bird’s-eye-view of the latest situation regarding GDM in India, we used the PubMed interface of Medline to search for ‘GDM and India’ during the last 5 years, 2016–2021. This study was not a comprehensive metanalysis of all the publications during this period. Two authors (UT, MMA) examined the search results. The other two authors (HD, RJ) also agreed with the studies selected. The authors, being from different geographical areas, and belonging to different diabetes and obstetric organizations, recommended some other classic studies and references, which are listed under the References section.

## 3. Results

There were 197 papers which met the search criteria. However, only 16 were deemed to be of use for this review. These articles are outlined in Table 1.

## 4. Discussion

The discussion is segregated into multiple separate headings for clarity.

## 5. Challenges Caused by Diabetes Mellitus

Though T2DM has become an epidemic across the world [23], it is the seven countries of SEA (Afghanistan, Bangladesh, Bhutan, India, Maldives, Nepal, Pakistan, and Sri Lanka) that bear the brunt, accounting for approximately a quarter of the global burden of diabetes and prediabetes [24]. In 2019, 88 million adults with DM were living in SEA, with over half the adults (57%) being undiagnosed [24]. If the current trends continue, 114 million Indians will be affected by 2045 [5] and India may overtake China as the DM capital of the world.

A recent study pooling 69 studies from 1,778,706 adults in India showed that between 1972 and 2019, the prevalence of T2DM increased in rural and urban India from 2.4% to 15% and from 3.3% to 19.0%, respectively [21]. Since GDM parallels the prevalence of T2DM, any increase in diabetes mellitus prevalence changes the number of women affected with GDM. Thus, the prevalence of GDM in Asia is higher than that in European countries as T2DM prevalence is much higher in Asians. Recent epidemiological data show that T2DM is affecting much younger women, who develop HIP once they become pregnant. The mortality and morbidity of DM and GDM would continue unabated without urgent action and preventive programs. However, since women constitute half of any population, and GDM is a marker for T2DM, active follow-up of women postpartum can help to contain the increasing prevalence of DM.

## 6. India: A Microcosm of Global GDM Trends

Worldwide, the approach to GDM has been diverse and chaotic without much agreement. In USA, the 1979 National Diabetes Data Group guidelines were amongst the earliest efforts towards an objective and scientific approach to DM and GDM [25]. This initial approach, which was followed by four international workshops on GDM (1979–1997), recommended major changes in the screening and diagnosis of GDM. Over the next two decades, the American Diabetes Association (ADA) and the American Association of Obstetricians and Gynecologists (ACOG) followed this trend by updating their guidelines for GDM and HIP; similarly, the World Health Organization (WHO) classifications of 1980, 1985, 1999 and 2013 have changed over time; and the National Institute for Health and Care Excellence (NICE) in United Kingdom, The Canadian Diabetes Association (CDA), and the Australian Diabetes in Pregnancy Society (ADIPS) were some of the major organizations (among others) of which the criteria were updated using local and global research. Therefore, due to tremendous global efforts, coupled with extensive research and international consensus conferences over the last six decades, the treatment of GDM shows much more agreement in terms of screening, diagnosis, management, and post-partum follow-up [26,27]. Thus, for instance, the segregation of GDM from DIP under the umbrella of HIP has clarified the definition and its relation to HIP is now more defined [28]. Furthermore, the evolution of laboratory technology and facilities and the acceptance of oral hypoglycemics has improved the screening, diagnosis, and treatment of GDM. The International Association of Diabetes and Pregnancy Study Groups (IADPSG) criteria from 2010 [29], using the 75-g oral glucose tolerance test (OGTT), have been adopted by many major organizations, such as the ADA, WHO, and ADIPS, among many others [30]. Some differences remain, and the current criteria using the 75-g OGTT are shown in Table 2. The 100-g OGTT is still recommended by the ACOG, with a 50-g glucose challenge test (GCT) screen.

GDM in India has been following the global trends. Like the rest of the world, the approach to GDM in India has been disorganized. Due to the myriad guidelines available, different algorithms were being followed, depending on the healthcare provider involved and the resources available. In general, densely populated countries with a high prevalence of T2DM make the implementation of any measure an overwhelming task; frequently, these nations have poor healthcare facilities, and inadequate economic resources. However, with the increasing prevalence of HIP, it is becoming even more critical to incorporate all the elements of GDM—screening, surveillance, management, and prevention—into every level of healthcare (primary, secondary, and tertiary). An equitable delivery of services in rural and urban areas would have a large impact on reducing the disease burden and preventing much of the morbidity and mortality associated with GDM, ante-partum and post-partum. Fortunately, as in most countries of the world, the GDM story in India has been evolving for the better.

## 7. Problems Involved in Managing GDM Specific to India

Every country faces an individual set of challenges in adhering to international guidelines, which, though scientifically sound, may be impossible to apply at the smaller Indian town- and village-level [31]. Implementing expert guidelines remains the weakest link in managing any disease [32]. Some obvious problems fall under three categories, namely, (a) the paucity of resources, such as trained manpower, phlebotomists, standardized equipment, and laboratory facilities [8], (b) the lack of information and motivation among caregivers (and patients) (a recent study showed that over 65% of health workers were not aware of GDM [22]), and (c) cultural problems, such as societal pressures, cultural beliefs, and dietary beliefs. Studies in the field have shown that pregnant women in India are afraid that the recommended diets will not provide ‘enough nourishment for the fetus’, and such diets may not be financially feasible. In India, insulin has been looked upon with trepidation, even when available without any cost to the patient. Culturally, exercise may be deemed to be inadvisable in pregnancy as rest is more important. The title of a recent paper is telling: *“If I don’t eat enough, I won’t be healthy”; Women’s experiences with gestational diabetes mellitus treatment in rural and urban South India* [17]. Other studies show that the challenges identified also include the following: screening women during the recommended time-period, testing women in the fasting state, scarcity of test consumables, lack of equipment, and the screening procedure being too time consuming [33]. Thus, researchers have tried to introduce the best scientific practices for GDM despite all the challenges—cultural and otherwise. The international body, the International Federation of Gynecology and Obstetrics (FIGO), advises modifying the approach to GDM to local settings; this is considered pragmatic and feasible by healthcare systems [31].

## 8. The Indian Healthcare System

Providing universal health care to nearly 1.4 billion people is a daunting task. Essentially, health care in India is provided by (a) public healthcare run by the government (federal and state) and (b) private healthcare (Figure 1).

### 8.1. Public Healthcare System

Public healthcare in India [8,34] is free of charge and therefore without any cost to the patient. It is provided primarily under the aegis of the federal Ministry of Health (MOH), and it is (using the local states) administered via a three-tier system (depending on the population) for the rural areas: (a) sub-centers (SCs) service populations of 3000–5000 people, and represent the first level of contact. Each is staffed with at least one auxiliary nurse midwife/female health worker and male health worker. These centers provide services and education for maternal health, immunization, and communicable diseases. (b) Primary health centers (PHCs) service populations with 20,000–30,000 people. A PHC has a medical doctor, paramedical staff and 4–6 inpatient hospital beds; it oversees up to six SCs, serving as their referral center. (c) Community health centers (CHCs) service 80,000–12,000 people. They have at least four medical specialists, including a gynecologist/obstetrician. There are laboratory and radiology facilities and 30 inpatient beds; these centers act as referral centers for PHCs. The MOH also directs additional sub-district, district and medical college hospitals in the urban areas, which provide secondary and tertiary care. Besides the MOH, the public health care is also provided by the Ministries of Defense and Railways; this care is limited for their employees and their families; however, being large employers, they service a substantial portion of the population. Their facilities consist of clinics and hospitals dotting the entire country.

In 2005, a national rural health mission (NRHM) added strength to the rural sector in terms of manpower and facilities, specifically stressing maternal and childcare. The MOH also runs other federal schemes, funding medical colleges, national health institutes, and tertiary care hospitals [8].

In 2018, the Government of India decided to start the Ayushman Bharat Program, which included two initiatives: (1) Health and Wellness Centers to upgrade over 150,000 PHCs to service the entire population, and (2) The Pradhan Mantri Jan Arogya Yojana to improve hospitalization (secondary and tertiary level) for the poorer strata of the population (the lower 40%).

### 8.2. Private Healthcare System

The private health care is for profit and takes the lion’s share of healthcare, especially in the urban areas. In cities and large towns, the first level of contact for patients are private clinics and nursing homes, often run by individual physicians or a group of physicians. Specialized hospitals with secondary, tertiary, and quaternary care are also present in urban areas and cities. The private healthcare system also includes non-profit, non-governmental organizations, which run free and philanthropic clinics. Furthermore, there are also charitable trusts which run dispensaries and hospitals in urban areas.

Thus, both public and private care follow a tier system, with super-specialty hospitals which are at the top of the totem pole for providing health care.

India has a long history of using alternative medicine, which has been used since antiquity. These forms of medicine are well-accepted by the public, as well as the Government of India, which established the Ministry of AYUSH, an acronym for ayurveda, yoga, unani, siddha, and homoeopathy [35]. This ministry was formed in November 2014 to ensure the optimal development and propagation of AYUSH systems of health care. However, we have avoided addressing the treatment offered by these systems of medicine. This study is limited to the use of allopathic medicine.

## 9. GDM in India

### 9.1. Prevalence of GDM in India

In any population, the prevalence of GDM depends on many factors in addition to the attributes of the cohort studied: maternal age, family history of diabetes, maternal weight and BMI, multiparity, and ethnicity. A meta-analysis of eighty-four studies showed that the pooled prevalence of GDM in Asia was 11.5% (95% CI 10.9–12.1) [11]. However, there is considerable heterogeneity in GDM prevalence, which was attributed to differences in diagnostic criteria, screening methods, and the study setting, e.g., urban vs rural, hospital vs community. Another study, using data from 51 population studies, showed that the prevalence of GDM is impacted by the different criteria used for diagnosis [36]. In a study on an Indian population, applying the diagnostic thresholds of eight major expert guidelines to the same glucose values of the diagnostic OGTT, the GDM prevalence varied from 9.2% to 45.3% depending on the criteria that were used [37]. Thus, the prevalence GDM in India has varied from 3.8% in Kashmir to 35% in Punjab [38]. In another review of 64 studies reporting 90 prevalence estimates, the prevalence of GDM in India varied from 0–41.9%; again, the authors blamed the differing criteria used in diagnosis for this variation [12].

Since differing criteria for GDM diagnosis have been blamed for the varying prevalence across India, to avoid the vagaries of different criteria, the same criteria (either random plasma glucose (RBG) ≥ 200 mg/dL or fasting glucose ≥92 mg/dL) were applied across the country by a major study published in 2020 [18]. The study cohort of 31,746 pregnant women (using 2015/2016 data from the National Family Health Survey of the Ministry of Health and Family Welfare) from 28 states and eight union territories of India demonstrated that high BMI (>27.5), older age, household wealth, woman’s caste and state of residence influenced the prevalence. The caste system in an archaic system practiced in India, in which the social status of a family was based on their birth. This system has been abolished by the Indian government and judiciary, but it persists in many areas. In essence, this study confirmed the wide variability in GDM prevalence across different geographic areas. The importance is that the same criteria (though epidemiological) were applied. However, the criteria used are not recommended by any scientific organization in the world. The authors do make a valid point: a varied (as opposed to ‘one-size-fits-all’) approach in different parts of India would be cost-effective.

### 9.2. Number of GDM Births in India

Every year, GDM afflicts between 5–8 million women in India [18]. The birth rate in India is approximately 25 million per year; this would translate into to one third to one quarter of births being affected by GDM. These figures are agreement with the 2019 International Diabetes Federation (IDF) Atlas (9th edition), which states that in SEA, one birth in four live births occurs in women with HIP compared to one birth out of six births globally [5]. In short, GDM is a bigger problem in India as it is far more prevalent.

Despite these challenges, it is encouraging to note that in Bangladesh, India, Pakistan, and Sri Lanka (the four largest countries, responsible for over 80% of the burden of HIP in SEA) have tackled these challenges: large-scale credible initiatives addressing barriers to treatment have been implemented. These initiatives, with the help of informed policy makers, generous federal and state funding, international agencies, and dedicated local care providers should make testing pregnant women for HIP as routine as the current and successful antenatal testing for anemia. Some of these measures in India are outlined in the remaining portion of this review.

### 9.3. The Beginnings of GDM Screening and Diagnosis in India

In the late 1990s, the Indian GDM story began to develop from its earlier rudimentary form. Until that time, for the screening and diagnosis of GDM, obstetricians were using whatever their set-up offered, and anything that was feasible. Usually, there were major variations in the approach of public and private sectors; thus, a pregnant woman visiting a public PHC may have been offered only (a) urine glucose or (b) random capillary glucose testing using a glucometer. During the same period, a patient in the private clinic/hospital would possibly be given (a) the then-recommended 2-step test for the screening and diagnosis of GDM (50-g GCT followed by a 3-hr 100-g OGTT) or (b) the 75-g OGTT as recommended by the WHO 1985/WHO 1999.

Therefore, the approach to GDM in India, as in many parts of the world, was plagued by the following problems: (1) there was no standard method for screening; (2) there was no consensus on whom, when, and how often to screening; (3) the paramedical staff were often poorly educated about GDM; and (4) there was poor awareness of the impact and long-term consequences of GDM.

As the worldwide understanding of GDM evolved, many questions were asked by Indian researchers and obstetricians. Was the ACOG/ADA-recommended standard 2-step screening even feasible in India especially on such a wide scale? There were problems in applying the global understanding of GDM screening into India. Field trials showed that many pregnant GCT-positive women were often difficult to trace to make them return for testing for a diagnostic GDM. Several villages were remote, with poor public transport systems. Thus, the recall rate was poor. The challenges did not end there—out of the women who did show up, many did not want to wait to complete the test. Many women could not be found after giving one or two blood samples to finish the remaining OGTT.

## 10. Further Progress in the Treatment of GDM in India

Due to the progress in GDM treatment globally, with the ease of access to knowledge, Indian health professionals have always been well aware of the overall health and economic benefits that accrue from identifying and treating GDM, both in the short term (perinatal complications) and the long term through postpartum lifestyle interventions to prevent or delay the onset of T2DM, obesity, and cardiovascular diseases, both in the mother and offspring. It was realized that the screening protocol in India had to be standardized by clearly defining (a) whom to test, (b) how to test, and (c) when to test.

### 10.1. Whom to Test? The Move from Selective to Universal Screening

In India, a technical assistance group (TAG) on DM within the Ministry of Health led to the development and publication of the GDM guidelines in 2014 by the government of India [39]. These recommended moving towards universal rather than selective clinical screening for GDM, which had been followed previously. Based on the data showing that Asians were 11 times more prone to GDM, FIGO, the Federation of Obstetric and Gynaecological Societies of India (FOGSI), and the Government of India recommended universal testing for all pregnant women. The experts within the country recognized that selective testing would translate into missed opportunities and that testing should be offered to all women irrespective of age, family history of DM, and previous obstetric history.

Thus, the concept that we must move from selective to universal screening was beginning to consolidate. However, more critically, a transition from awareness to practical initiatives was instituted.

### 10.2. How to Test? The DIPSI Test

The next big question was which universal screening test was practicable for all the pregnant women in India? Was it the 50-g GCT? The full 75-g OGTT? Any form of screening with 25 million births annually in India would place a huge burden on a poorly resourced health system. Any recommendation for testing women for HIP would therefore need to be pragmatic, feasible, convenient, and cost-effective. Another problem in India was collecting a fasting glucose sample for multiple reasons. The cultural belief that pregnant women should not fast for long hours was a major problem, insular to India. Furthermore, the drop-out rate became higher if the pregnant woman was asked to come again the next day (after the routine obstetric visit) for an OGTT.

One solution was proposed after a landmark study by Seshiah et al. which claimed that a plasma glucose 2-h value after 75-g OGTT ≥ 140 mg/dL was comparable to the GCT and the 75-g OGTT which used both fasting and the 2-h plasma glucose value [40]. In essence, this test was modified from the now obsolete WHO-1999 criteria for 75-g OGTT to diagnose GDM, but was performed in a non-fasting state. This test was potentially usable for many reasons: it was simple to perform, economical, it did not need a fasting glucose sample, and compared favorably with more elaborate tests of glucose intolerance in pregnancy. The authors touted this test with an Albert Einstein quote, “Most complicated problems in the universe have a simple solution”. Thus, in 2006, the Diabetes in Pregnancy Study Group, India (DIPSI) decided that venous plasma glucose should be measured (despite a non-fasting state, irrespective of the last meal) 2 h after the 75-g glucose test. This test was better known by its acronym: the DIPSI test. Subsequent studies [41] confirmed that the study by Seshiah et al. correlated well with the WHO 1999 recommendations, as it was similar but was performed in a non-fasting state; however, some studies did not agree with these findings [42].

### 10.3. National Guidelines for the Diagnosis and Management of Gestational Diabetes Mellitus, India

In 2014, the Maternal Health Division of the Ministry of Health and Family Welfare, Government of India (MOHFW) published the national guidelines for the diagnosis and management of GDM in India [39]. They recommended the DIPSI test at booking, and if it was negative, to repeat it between 24–28 weeks gestation for all pregnant women. A standardized glucometer was to be used, and it needed to be calibrated using a standard protocol. They updated their guidelines in 2018 [43], using national and international studies. Experts studied the logistic limitations and stressed newer technical operative ideas. As in the first recommendations, testing a woman two times remained central to the updated guidelines [13]. However, they stressed the proper use of operational guidelines, using the glucometer for self-monitoring (SMBG) and the use of oral hypoglycemic agents (OHA), as outlined below.

(A) Point of care using a glucometer: ideally, for the diagnosis of GDM, test results should be based on venous plasma samples, collected properly and transported quickly for testing to the laboratory, which must be accredited by professional laboratory bodies. However, this ideal situation is difficult to attain, especially in many primary care settings, particularly in PHCs in remote areas—where proper facilities for collection, transport, storage, or testing may not exist. In this situation, it was acceptable to use a plasma-calibrated hand-held glucometer with properly stored test strips to measure plasma glucose. Using a glucose meter in this situation may be more reliable than laboratory tests done on samples that were inadequately handled and poorly transported. Using a glucometer has made screening possible in many remote areas where it was not possible earlier. However, proper training of the staff was the key to success with a laboratory glucometer.

(B) Self-monitoring-blood-glucose (SMBG): glucometers were also used widely and effectively for the monitoring of blood glucose once GDM had been diagnosed. Many women preferred a daily blood glucose check rather than a weekly/fortnightly fasting and postprandial check in the laboratory. By using a glucometer and testing herself often, the patient was more involved in her treatment. Furthermore, she realized which foods to avoid, and she was able to inform clinicians about her glucose control.

(C) The use of oral hypoglycemics (OH): metformin has been approved for use in GDM after 20 weeks of gestation by the Central Drugs Standard Control Organization of India, which is the pharmaceutical watchdog, equivalent to the US Food and Drug Administration (FDA). The advantages of OH are as follows: it is easy to take, being an oral medication; it seldom causes hypoglycemia; it is more readily acceptable to the patient (compared to insulin); and it is much cheaper. It also limits the use of insulin to a select few women—those who cannot tolerate metformin or women in whom metformin fails to control the plasma glucose.

## 11. Criticisms of the DIPSI Test

As mentioned earlier, the DIPSI test was approved by the Diabetes in Pregnancy Study Group India and the MOHFW, Government of India. This test seemed to be the only choice for India because it was a pragmatic, single test, which was a walk-in test, with just one glucose value needed to diagnose HIP [15].

However, it has its critics. One of the arguments made against it is that the landmark hyperglycemia and pregnancy outcome (HAPO) study showed that higher isolated fasting glucose levels are related to the higher incidence of the occurrence of poor maternal and fetal outcomes, such as fetal hyperglycemia, future diabetes, premature delivery, intensive neonatal care, hyperbilirubinemia, preeclampsia, shoulder dystocia, or birth injury [44]. Many studies have studied the DIPSI test, as follows:In a cohort of 152 pregnant women, the DIPSI and the IADPSG protocols were compared; 34 (22.4%) women, i.e., one in five, were missed by DIPSI compared to IADPSG [45]. Since IADPSG is the most popular world guideline, accepted by many major organizations such as the WHO 2013 and ADIPS, this study has many serious implications. Though it was a small study, it has serious implications.Another prospective study involved 936 pregnant women, who underwent plasma glucose evaluations two hours after the challenge of 75-g glucose load, irrespective of the timing of their last meal (DIPSI criteria for GDM). After three days, standard 75-g OGTT anlysis was carried out for all women. When compared with the IADPSG criteria, the sensitivity was 74.1 percent, and the corresponding specificity was 96.9 percent [9]. Again, this study impliedthat the DIPSI test missed about 16.0% of women.A prospective study on 200 women aimed to validate the single-step non-fasting 75-g DIPSI criteria of GDM in Indian patients in comparison with the two-step fasting 100 g OGTT using the Carpenter–Coustan criteria (CCC). All 200 patients were subjected to comparative testing through a non-fasting 75-g oral glucose (DIPSI) and fasting 100-g OGTT, interpreted using CCC at less than 20 weeks POG and again between 24–28 weeks POG. The DIPSI compared well with the CCC criteria [14].Another study showed that of 106 women picked up by IADPSG, only 24 (22.6%) women showed GDM as determined by the DIPSI [42].In a study on 274 women in Sri Lanka, a non-fasting, 75-g, non-fasting DIPSI test was followed by the standard 75-g fasting OGTT using IADPSG criteria. However, the sensitivity and specificity of the DIPSI were poor, at 40.6% and 94.4% respectively [46]. About half the women would have been identified by fasting plasma glucose alone, which is omitted by the DIPSI.


Thus, the DIPI test did not seem sensitive enough. A review of screening tests for gestational diabetes in South Asia from 19 studies were evaluated. The authors found that the DIPSI had better specificity than sensitivity and thus recommend that it should be used as a diagnostic test rather than as a screening test [19]. Despite much criticism, this single-step test remains the only feasible test for large populations in low- and middle-income countries (LMIC).

## 12. Initiatives for Managing GDM in India

### 12.1. Collaboration of the Federation of Obstetric and Gynaecological Societies of India (FOGSI) and the Govenment of India

In 2016, FOGSI initiated a certification course (online and offline) called Manyata [47]—a diabetes screening protocol in antenatal care. It covered (a) the capacity building of healthcare personnel to improve the quality of diagnosis and treatment, and (b) the training of health care personnel about GDM screening/treatment, nutrition, and data collection. Manyata was a quality improvement and certification initiative designed to ensure that all private facilities met quality standards by building health care workers’ capacity and awarding a certificate.

Workshops were held by FOGSI, FIGO, and DIPSI across the length and breadth of India—from metropolitan areas to smaller towns to spread this awareness. FOGSI members led these continuing medical education initiatives, teaching, clarifying the doubts of practitioners, and spreading awareness not only on screening, but also on managing GDM, as well as stressing the importance of follow-up.

In 2017, a survey of the teaching faculty of the obstetrics and gynecology departments of 47 medical college institutions in India revealed that almost all (95.8%) participants reported that all pregnant women were offered (universal) testing for hyperglycemia in pregnancy [10]. Most (70.21%) respondents used the single-step, non-fasting method to diagnose HIP. Most respondents (89.4%) reported having noticed an increase in the number of women with hyperglycemia. A crucial finding was that over nine in 10 of all the respondents felt that there was a need to train medical personnel to test and manage hyperglycemia. Two-thirds of the respondents felt that all women readily agreed to undertake the DIPSI test, and it was not a challenge to convince them to undergo the test. This study confirmed that the increasing adherence to and awareness of national guidelines has the potential to result in the earlier diagnosis and management of HIP. This would improve pregnancy outcomes, maternal, and neonatal health in the short term as well as in the long term.

### 12.2. The International Diabetes Federation (IDF) Approach for GDM in India

The Women in India with GDM Strategy (WINGS project) is one of IDF’s flagship projects. It aimed to find a model of care (MOC) for low-resourced settings [20]. It started in 2012 and had four phases. It addressed problems at every level—the individual, family, health system, and community levels—and published its results in 11 scientific articles. It used a multi-level approach: flyers and posters, using games as educational tools, field visits, awareness activities, press meetings, and healthy cooking and diet tutorials. The result was that women with GDM had outcomes comparable to women without GDM, which was a real breakthrough.

### 12.3. The World Diabetes Federation–Jagran–Medanta Project

Since public awareness is one key factor in GDM management, a mass information program was started in 2010–2011. Jagran Pehel (a social initiative), along with WDF and Medanta, an India chain of multi-specialty hospitals, has worked to increase awareness of GDM. Using mobile phones, web-based forums, print media, and radio campaigns, over half a million people were reached. The experience from such projects is vital for state and federal run projects. Ultimately, helping to disseminate information about GDM to the masses is crucial in clearing cultural and other roadblocks. The WDF has also been working on many other projects in India [48].

## 13. The Post-Partum Challenges and Lifestyle Intervention

Actively pursuing women with GDM after delivery can help to contain the epidemic of T2DM through preventive strategies [4]; however, most women with GDM are lost in follow-up after delivery. In a study of women with GDM after delivery, a control group in India [16] with strict postpartum follow-up for 17 months (mean) was completed with six sessions on lifestyle modifications, carried out in four groups of 12–15 women each. All indices of metabolic health improved (weight, glycemia, lipids, and BP) and 70% of women with prediabetes postpartum reverted to normoglycemia. Though this study proved that intervention after delivery works, translating it into success in the entire country remains a challenge.

The 2018 government of India guidelines stress the importance of counselling about lifestyle modifications, weight control, exercise, and family planning. A post-partum 75-g OGTT was recommended 6 weeks after delivery.

The post-partum follow-up of women with GDM with strict preventive and lifestyle modifications remains important in order to turn the tide of the epidemic of T2DM. However, post-partum follow-up continues to be the weakest link in managing GDM, which is a pattern seen even in the most advanced countries in the world.

## 14. GDM Testing during the Pandemic

The COVID-19 pandemic forced a rethink about the ideal way to approach HIP. The IADPSG test involves visiting the laboratory and waiting for 2 h, thereby increasing the risk of pregnant woman being exposed to COVID. Realizing this, many countries have changed their guidelines to a one-step process—fasting or HbA1c—though both tests are not sensitive enough in picking up GDM [49]. DIPSI test, which is already an established method, could be done at home using a glucometer. Thus, while the world has struggled in experimenting with how to modify screening for GDM during the COVID era, in India it has been business as usual because DIPSI is a simple and pragmatic test used to screen for pregnant women with GDM.

## 15. Conclusions

India represents a microcosm of the great strides made in achieving healthy HIP pregnancies globally. The importance of managing GDM is simple; if it is diagnosed, treated, and followed up after delivery, it will bring down a country’s load of T2DM in the current generation, as well as in the next generation. Thus, GDM follow-up has great potential for positively changing the future predictions in relation to T2DM.

It is well known that 90% of HIP births occur in the LMIC. Great challenges exist in implementing universal screening for GDM, with coverage varying between 10% and 90%. Missing and not treating women with GDM will increase the immediate and long-term complications associated with GDM. India shares challenges common to all countries with limited resources: how to implement the latest research despite constraints of manpower and laboratory consumables and bureaucratic roadblocks. There are also major cultural differences, which create logistical issues. Currently, the IADPSG screening method has the largest following globally, but is doable only in small pockets in India as, as shown by the International Diabetes Federation’s Women in India with GDM Strategy (IDF-WINGS) study [20]. However, this approach may not be currently feasible in India. Due to the diversity of GDM prevalence in different states of India and given the country’s gigantic size, a method tailored to the burden of disease to maximize cost effectiveness may be the answer [14]. Experts agree that one size may not fit all, and it may be best to adapt solutions to local situations [20], a sentiment that is also echoed by FIGO [31]. The DIPSI test remains popular, and despite its criticisms, it is widely accepted by major local organizations. Currently, it appears to be one acceptable potential method. Clearly, there are no easy solutions.

A multi-pronged approach, using extensive educational campaigns, the effective use of resources (including federal, state, international, and philanthropic organizations) may also provide significant assistance. Currently, such efforts are ongoing. The caretakers of pregnant women look to the experts for guidance. These preeminent researchers have always been supportive; however, we need them to provide one clear guideline. It is a matter of time before the screening, diagnosis, management, and—most crucially—the post-partum follow-up of GDM in India become as good as those of any country in the world.

## Figures and Tables

**Figure 1 medicina-57-00942-f001:**
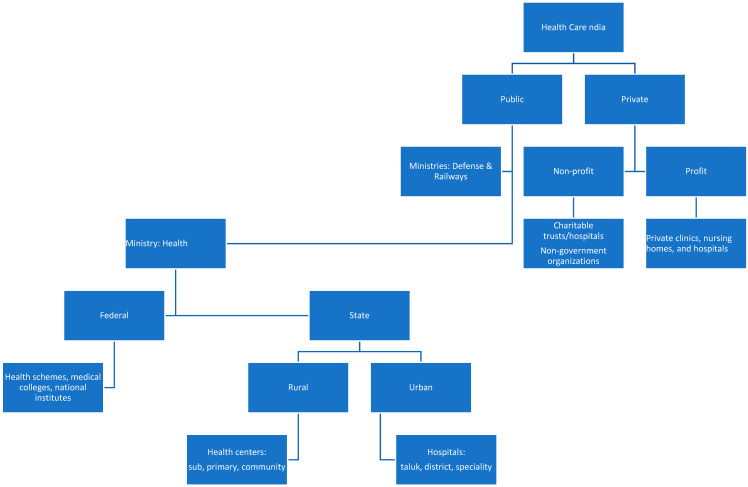
Healthcare in India: basic structure.

**Table 1 medicina-57-00942-t001:** Studies of gestational diabetes (GDM) in India included for the last 5 years (2017–2021).

Study (Reference)	Year	Topic Addressed	Article Type
Morampudi et al. [8]	2017	GDM care in India	Review
Tripathi et al. [9]	2017	Evaluation of DIPSI test	Study
Devakar et al. [10]	2017	GDM practice in medical schools	Study
Lee et al. [11]	2018	Prevalence of GDM	Review
Li et al. [12]	2018	Screening and diagnosis of GDM	Review
Mishra et al. [13]	2018	GDM guidelines in India	Review
Khan et al. [14]	2018	Evaluation of DIPSI test	Study
Seshiah et al. [15]	2019	Diagnostic criteria of GDM	Review
Kapoor et al. [16]	2019	Effect of lifestyle changes on GDM	Study
Kragelund et al. [17]	2020	Women’s understanding of GDM	Study
Swaminathan et al. [18]	2020	Prevalence of GDM across India	Study
Lappharat et al. [19]	2020	Screening tests for GDM	Review
IDF WINGS [20]	2020	Women in India with GDM strategy (WINGS) project of the International Diabetes Federation (IDF)	Review
Swaminathan [18]	2020	GDM prevalence across India	Review
Ranasinghe et al. [21]	2021	Trends of diabetes in rural and urban India	Review
Archita et al. [22]	2021	GDM perception of healthcare workers	Study

**Table 2 medicina-57-00942-t002:** Current diagnostic criteria of GDM with diagnostic values for the 75-g oral glucose tolerance test.

Criteria	Year	Abnormal Glucose Values Needed for Diagnosis	Plasma Glucose Thresholds mmol/L		
			0-h	1-h	2-h
IADPSG/WHO	2010/2013	≥1	5.1	10.0	8.5
CDA	2013	≥1	5.3	10.6	9.0
NICE	2015	≥1	5.6	--	7.8
ADA (C&C)	2003 (1982)	≥2	5.3	10.0	8.6

IADPSG, International Association of Diabetes and Pregnancy Study Groups; CDA, Canadian Diabetes Association; WHO, World Health Organization; NICE, National Institute for Health and Care Excellence; ADA, American Diabetes Association; C&C, Carpenter and Coustan.

## Data Availability

All the studies used in this review are referenced.

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
