# Peer review of "Negotiating Gestational Diabetes Mellitus in India: A National Approach"

_medicina, 2021, doi:10.3390/medicina57090942_

Round 1

Reviewer 1 Report

The manuscript offers an overview of diabetes mellitus and gestational diabetes mellitus in India from public health point of view. 

Here are some critical remarks that I find important:

  1. The manuscript drifts asway from the title somehow as not much is said about prevention.
  2. The abstract does not really confer the manuscript content.
  3. The authors' choice of words in some parts of the manuscript is interesting but not strictly scientific and in some instances more appropriate for a press release. Some examples: ivory-tower background; myriad roadblocks indigenous...; current pestilence; grassroot level; taken the bull by the horns... etc.
  4. Some portions of the manuscript have the sounding of an administrative report rather than a scientific paper.
  5. Some statements create an impression of a possible causality: lines 82-84: higher GDM rates do not result from higher MD2 prevalence. It's the opposite. 
  6. There are few data discrepancies: The birth rtate is said to be 25 million/year at line 252 and 37 million at line 322.

Author Response

The manuscript offers an overview of diabetes mellitus and gestational diabetes mellitus in India from public health point of view. 

Here are some critical remarks that I find important:

  1. The manuscript drifts asway from the title somehow as not much is said about prevention.

Reply: We fully agree. Thank you for this great insight. We tried to stress that education of the masses (through home visits, pamphlets, working at ground level) has helped to in preventing the complications in index pregnancy. Also, better follow-up after delivery has helped to delay the impending T2DM.   However, this manuscript was written for an international audience, so, we had to give an insight into the health care system in India. Similarly, we had to use broader historical insights to explain the evolution of GDM in the country. Thus, we had to stray away from the title in strict terms. Also, we tried to highlight that a large heterogenous country can still attain success –despite all the constraints.

We have modified the title to -- Negotiating gestational diabetes in India. This captures the manuscript better. We hope you like it better. Thank you for this comment.

2.The abstract does not really confer the manuscript content.

Reply: We fully agree. However, due to the multiple, disparate aspects (health care system in India, health delivery, multiple approaches to GDM) covered in the manuscript, we found it onerous (if not impossible) to distill the manuscript into a succinct abstract. Thus, we provided a general overview in the abstract, enough to whet the appetite for the reader, potentially motivating them to read the manuscript. We have made a few changes in language and it reads much better.

We do appreciate your comments. Thank you.

3. The authors' choice of words in some parts of the manuscript is interesting but not strictly scientific and in some instances more appropriate for a press release. Some examples: ivory-tower background; myriad roadblocks indigenous...; current pestilence; grassroot level; taken the bull by the horns... etc.

Reply: Thank you. The language reflects the authors’ upbringing and the syntax and vocabulary used. We agree that it may sound more like a newspaper report. However, this was not strictly a scientific or a meta-analysis –as we point out. We view it more as an informal informative review about the Indian battle against GDM. We have used alternate words for all the vocabulary that you mention. We would be happy to change any others. Ultimately, any comment by our reviewers makes the article more readable to the readers of the journal.

4. Some portions of the manuscript have the sounding of an administrative report rather than a scientific paper.

Reply: We could not agree more. We have changed some of the syntax what you point out in your constructive comment #3, earlier. Thank you.

5. Some statements create an impression of a possible causality: lines 82-84: higher GDM rates do not result from higher MD2 prevalence. It's the opposite. 

Reply: Apologies. Again, you are correct. We have rephrased it.

6. There are few data discrepancies: The birth rate is said to be 25 million/year at line 252 and 37 million at line 322.

Reply: We checked various sites and the birth rate in India is about 24.1 million per year. Thank you for this pick-up. We acknowledge the error ---and thank you.

We appreciate all your time and effort. All your comments have been very constructive. We shall make any other changes that you feel would improve our manuscript. Thank you, again.

Reviewer 2 Report

The article provides a comprehensive review of the very interesting topic of gestational diabetes prevention in both developed and less developed countries. The article focuses on India, where both prevention and intervention strategies have been put in place to limit the impact of this health problem.

The main strength of the study is that it does a very important job of compiling interventions in a specific country such as India where gestational diabetes is a major health problem. It also indicates a large number of studies conducted in this country, which provides valuable information.

As the most important limitation, and derived from what has been said above, the data obtained cannot be extrapolated to other countries, which complicates the comparison of results with those obtained by other researchers.

The keywords Under-Developed Countries and Preventive Health Programs are not in the Mesh, we suggest they be changed by: Less-Developed Countries and Prevention

As specific data to be modified, we would like to point out the absence of the year in references 15 and 16.

The obsolescence index of the references is 4 years, which is very appropriate for an article of these characteristics, however almost half of the citations are more than 5 years old and more than 20% are more than 10 years old, so we recommend updating some citations, especially those that are more than 10 years old.

Author Response

The article provides a comprehensive review of the very interesting topic of gestational diabetes prevention in both developed and less developed countries. The article focuses on India, where both prevention and intervention strategies have been put in place to limit the impact of this health problem.

Reply: You summarized the intent of our paper succinctly. We hope that our message comes out loud and clear. Thank you for this comment.

The main strength of the study is that it does a very important job of compiling interventions in a specific country such as India where gestational diabetes is a major health problem. It also indicates a large number of studies conducted in this country, which provides valuable information.

Reply: We appreciate your positive feedback. The studies have been disparate, and to interpret them logically has been difficult, but we have tried. Thank you.

As the most important limitation, and derived from what has been said above, the data obtained cannot be extrapolated to other countries, which complicates the comparison of results with those obtained by other researchers.

Reply: Absolutely. We point out that the solutions have to be indigenous. The interventions succeeding in India, may fail in Sri Lanka. We have some experience working with Nigerian colleagues. The problems were quite different due to the culture. As we point out, in India, eating less during pregnancy is a taboo.

The keywords Under-Developed Countries and Preventive Health Programs are not in the Mesh, we suggest they be changed by: Less-Developed Countries and Prevention

Reply: We have made these changes. You are right. We have changed the key words. Thank you.

As specific data to be modified, we would like to point out the absence of the year in references 15 and 16.

Reply: We have corrected these errors. Thank you.

We do appreciate all your time and effort.
